

**The UNAM-MARine Aerosol Tank (UNAM-MARAT): An Evaluation of the**
**Ice-Nucleating Abilities of seawater from the Gulf of Mexico and the Mexican**
**Pacific**
M. Fernanda Córdoba[1], Rachel Chang[2], Harry Alvarez-Ospina[3], Aramis Olivos[4] Graciela
B. Raga[1], Daniel Rosas-Ramírez[5], Guadalupe Campos[6], Isabel Márquez[3], Telma Castro[1],
and Luis A. Ladino[1,*]
[1]Instituto de Ciencias de la Atmósfera y Cambio Climático, Universidad Nacional
Autónoma de México, Ciudad de México, C.P. 04510, México
[2]Department of Physics and Atmospheric Science, Dalhousie University, C.P. B3H 4R2,
Canada
[3]Facultad de Ciencias, Universidad Nacional Autónoma de México, México City, México
[4]Centro Universitario de Investigaciones Oceanológicas, Universidad de Colima, C.P.
28860, México
[5]Departamento de Química de Biomacromoléculas, Instituto de Química, Universidad
Nacional Autónoma de México, Av. Universidad 3000, Circuito Exterior S/N, Coyoacán,
Ciudad Universitaria, Mexico City, 04510, México
[6]Laboratorio de Alimento Vivo, Procuraduría Estatal de Protección al Medio Ambiente-
Aquarium del Puerto de Veracruz, Blvd. Manuel Ávila Camacho s/n, Col. Ricardo Flores
Magón, C.P. 91900. Veracruz, Veracruz, México
*Corresponding author: luis.ladino@atmosfera.unam.mx
Keywords: Ice Nucleating Particles, Sea Spray, Mexican Pacific Ocean, Gulf of Mexico.
**Abstract**
Although several studies have shown that sea spray aerosol (SSA) has the potential to act as
ice nucleating particles (INP) impacting cloud formation, there is a lack of marine INP studies
in tropical latitudes. This is partly due to the unavailability of local oceanographic cruises
that perform aerosol-cloud interaction studies in the tropics, as well as the scarcity of
appropriate aerosol and cloud microphysics instrumentation. The present study shows the
development of the UNAM-MARine Aerosol Tank (UNAM-MARAT), a device that
simulates wave breaking to generate SSA particles with the main purpose to characterize
their physicochemical properties including their ice nucleating abilities. The UNAM-
MARAT was characterized using Instant Ocean Sea Salt and its potential to study ambient
sea waters was evaluated with sea seawater samples collected from the Port of Veracruz
(PoV) in the Gulf of Mexico, and from the Bay of Acapulco (BoA) and the Bay of Santiago-





Manzanillo (BoSM) in the Mexican Pacific Ocean. The portable and automatic UNAM-
MARAT is able to generate aerosol particle concentrations as high as 2000 cm$^{-3}$ covering a
wide range of sizes, from 30 nm to 10 µm, similar to those found in the ambient marine
boundary layer. The SSA generated from the three natural seawater samples was found to act
as INP via immersion freezing, with INP concentrations as high as 130.7 L$^{-1}$. The particles
generated from the BoA seawater samples were the most efficient INPs, reporting the highest
ice active site density ($n_s$) values between -20 and -30°C. Our results also show the direct
relationship between particle size and its composition. Larger particles (> 1 µm) were found
to be enriched in sodium chloride. In contrast, the fraction of $Ca^{2+}$, $Mg^{2+}$, and $NO_3^-$ was found
to increase with decreasing the particle size from 10 µm to 320 nm. This suggests the
presence of dissolved organic material in the submicron particles.

**1 Introduction**

Sea-Spray Aerosol (SSA) is ubiquitous in oceanic regions and forms via bubble bursting by
wave breaking (Lamarre and Melville, 1991). It has been shown that SSA has the potential
to impact the Earth's radiative balance (Jacobson, 2001) and the hydrological cycle given
its capability to act as cloud condensation nuclei (CCN, Albrecht, 1989) and ice nucleating
particles (INP, Boucher et al., 2013; Vergara-Temprado et al., 2017; McCluskey et al.,
52   2018).

Laboratory experiments with diverse setups, including atomizers, nebulizers, and tanks (in
acrylic, PTFE or stainless steel), to simulate SSA generation via bubble bursting, have been
essential in determining the physicochemical and biological properties of SSA (Fuentes et
al., 2010; McCluskey et al., 2017; Christiansen et al., 2019; Wolf et al., 2020). Some of these
setups used different mechanisms for bubble production such as diffusers, glass frits or
systems like plunging-water jet or sheetlike (Cipriano & Blanchard, 1981; Fuentes et al.,
2010; Prather et al., 2013; Stokes et al., 2013; Christiansen et al., 2019). Using a small tank
to produce SSA, Cipriano & Blanchard (1981) determined that bubbles with diameter > 1
mm can produce aerosol particles < 5 µm in diameter, while bubbles < 1 mm generate aerosol
particles > 20 µm. However, it is currently believed that submicron (< 1 µm) and supermicron
(> 1 µm) aerosol particles can be generated by the film drop and jet drop mechanisms,
respectively  (Resch & Afeti, 1992; Lewis & Schwartz, 2004; Burrows et al., 2014).
Recently, Wang et al. (2017) found that the jet drop mechanism can produce up to 43% of
submicron SSA.
Results from field measurements and laboratory experiments indicate that SSA exhibits a
trimodal particle size distribution (PSD) with peaks observed at 0.02 - 0.05 µm,  0.1 - 0.2
µm, and 2 - 3 µm (Quinn et al., 2015). Laboratory experiments using artificial seawater in a
30 L marine aerosol tank, Sellegri et al. (2006) demonstrated that the trimodal PSD of the
SSA can vary with other environmental variables such as sea surface temperature (SST). The
authors found that if SST decreases the peaks of the PSD are displaced towards smaller





diameters. The presence of surfactants (e.g., sodium dodecyl sulphate, SDS) can also
influence in the amplitude of the modal peaks as surfactants extend the bubble lifetime at the
surface, and then bubbles can be broken by wind or subsequent waves (Sellegri et al., 2006).
Additionally, Hartery et al. (2022) found that adding sodium dodecyl benzene sulfonate,
(SDBS, a surfactant) to a NaCl solution in the Dalhousie Automated Wave Tank (DAWT)
reduced particle size mode, particle concentration, and hygroscopicity, further highlighting
the impact of surfactants on aerosol properties. Using a similar experimental setup to the one
used by Sellegri et al. (2006), Fuentes et al. (2010) found that the SSA submicron size
distribution, its hygroscopicity, and its ability to act as CCN are not significantly affected by
the bubble bursting generation mechanism  (i.e., porous bubblers and plunging- water jet
systems). Nevertheless, Fuentes et al. (2010) reported that the best system for SSA generation
when using natural sea water was the plunging-water jet, which improves the reproduction
of organic enrichment and PSD.
Stokes et al. (2013) implemented a new system for SSA generation that includes an
intermittent plunging sheet of water in a plexiglass 210 L tank, called the Marine Aerosol
Reference Tank (MART). This mechanism simulates the gravitational impingement of a
waterfall and the intermittence better reproduces wave breaking to create turbulence, the
bubble plumes, and foam formation. The interaction of freshly emitted SSA with volatile
organic compounds present in the marine atmosphere has been evaluated in the MART.
Trueblood et al. (2019) discovered that by exposing supermicron SSA to hydroxyl radicals
(OH), a fragmentation of the nitrogen-rich species (e.g., amino sugars or amino acids) is
observed, and therefore, there is a reduction in the organic matter present in the SSA.
In addition to the above-mentioned laboratory tanks, a large-scale experimental setup such
as the Wave Channel have provided insights into SSA generation under realistic marine
conditions. A large tank (33 m x 0.5 m x 1 m) was designed in the Hydraulics Laboratory at
the Scripps Institution of Oceanography (SIO) in San Diego – United States (Collins et al.,
2014). SSA is generated through a hydraulic-paddle- created waves, sintered glass filters,
and an intermittent plunging sheet of water (Collins et al., 2014). Simulation of ocean
dynamics and biological activity in the wave channel allowed Prather et al. (2013) to
conclude that SSA is composed mainly of four types of particles: sea salt (SS), sea salt with
organic carbon (SS-OC), organic carbon (OC), and biological (Bio) particles, with its
chemical composition strongly linked to particle size. The authors reported that supermicron
particles were dominated by SS and Bio, while submicron particles by SS-OC and OC.
Prather et al. (2013) also reported that Na, Cl, Mg, and K largely contribute to the SS particles
and that between 30 and 40% (v/v) of the SS-OC particles, correspond to organic matter. Ca
and Mg were found to be present in the OC particles, and they are known to be able to form
complexes with natural organic ligands (Quinn et al., 2015). Organic matter can accumulate
in the air-ocean interface, forming a gel-like layer with properties that differ from the
underlying waters. This layer, known as the sea surface microlayer (SML), has a typical





thickness that varies between 1 and 1000 µm (Wurl et al., 2017). In an experiment similar to
that carried out in the wave channel by Prather et al. (2013), Wang et al. (2015) determined
that marine submicron particles are enriched in aliphatic organic material and that the soluble
oxidized organic compounds are found in supermicron particles. Additionally, Wang et al.
(2015) showed that differences in the SSA chemical composition could result from a variety
of biological processes, including bacterial activity and phytoplankton primary production.
It is well known that SSA can act as an INP (Bigg, 1973; Schnell & Vali, 1975; Schnell,
1977; Rosinski et al., 1987; Rosinski et al., 1988; Wilson et al., 2015;  McCluskey et al.,
2018). Several studies have suggested that marine biological particles are able to nucleate ice
such as the *Heterocapsa niei* (dinoflagellate) (Fall and Schnell, 1985) and *Thalassiosira*
*pseudonana* (diatom, Knopf et al., 2011; Alpert et al., 2011; Wilson et al., 2015). Through
controlled laboratory experiments in the MART, DeMott et al. (2016) demonstrated that the
INP number concentrations from seawater collected close to SIO - Pacific Ocean off the
California coast, are within the range reported by previous studies in different maritime
regions (Bigg, 1973;  Schnell, 1977; Rosinski et al., 1988). However, the INP concentrations
were lower than the corresponding concentrations in the surface boundary layer over
continental regions. DeMott et al. (2016) also noted that the INP concentrations at -26 and -
30°C were a factor of 50 larger after nutrient addition than freshly collected seawater. Wang
et al. (2015) found that maximum concentrations of INP at ≥ -15ºC coincided with the peaks
of the phytoplankton bloom carried out in the wave channel. These experiments used
seawater from the Pacific Ocean near the SIO, suggesting that the observed ability to act as
INP could be due to amphiphilic long-chain alcohols monolayer and that the ice nucleating
activity (INA) was reduced when the samples went through the heating test, a process to
denature biological INP (Hill et al., 2016). McCluskey et al. (2017) used seawater collected
from the Pacific Ocean at the end of Scripps Pier (32°49′58.12″ N, -117°16′16.58″ W) in the
MART. Their study suggests that microorganisms and biomolecules contribute to the INP
population due to an increase of organic compounds during high INP concentrations.
Studies on INPs along the Mexican coasts and oceans are scarce. A pioneering study by
Rosinski et al. (1988) in the Gulf of Mexico (GoM) demonstrated that the efficiency of
aerosol particles as INP varies depending on their size, season, and sampling location. The
influence of environmental conditions on the ice nucleation efficiency of marine aerosol
particles was also evidenced by Ladino et al. (2019) and Córdoba et al. (2021). Both studies
found that the arrival of cold air masses to the Yucatan Peninsula (Mexico) from higher
latitudes increased the INP concentrations with aerosol particles capable to nucleate ice at -
3°C.The warm freezing temperature suggests the influence of biological material, likely
linked to bacteria and fungi from terrestrial and/or marine sources. Although the analyzed
samples were not airborne particles, Ladino et al. (2022) found that the sea subsurface water
(SSW) samples from the GoM exhibited better ice nucleation abilities than the sea surface
microlayer  (SML) samples, contrary to the findings of Wilson et al. (2015) at higher



latitudes. This discrepancy could be attributed to a lower organic material content in the SML
samples of the GoM compared to those analyzed by Wilson et al. (2015). This difference in
nucleation efficiency was also associated with low phytoplankton concentrations during the
sampling period in the GoM, a crucial variable in the efficiency of particles as INPs.
Investigating SSA's role in marine environments is imperative for improving the accuracy of
climate predictions (Burrows et al., 2022). Given that SSA ability to act as INP varies
spatially and temporally  (Burrows et al., 2013; Wilson et al., 2015; DeMott et al., 2016) and
the scarcity of ice nucleation studies in tropical latitudes over maritime regions (Rosinski et
al., 1988; Yakobi-Hancock et al., 2014; Wolf et al., 2020; Córdoba et al., 2021; Ladino et al.,
2022; Melchum et al., 2023), expanding research efforts to study unexplored regions, such
as the Mexican coasts, is of high importance. By advancing our understanding of SSA
dynamics, we can enhance the accuracy of atmospheric models and reduce the uncertainties
associated with aerosol-cloud interactions, thereby, contributing to more robust climate
projections. The present study involves the building and characterization of a new device to
generate SSA by simulating wave breaking through the intermittent plunging sheet of water
mechanism, utilizing water samples from seawater collected offshore the Mexican coasts.
**2 Instrument development**
**2.1 Description and Operation of the UNAM-MARAT**
The UNAM - MARine Aerosol Tank (UNAM-MARAT) was built based on the design of
Stokes et al. (2013) to study the physicochemical properties of SSA and its ability to nucleate
ice under controlled conditions in the laboratory, using samples obtained from the oceanic
waters that surround Mexico.
The UNAM-MARAT consists of an acrylic tank of 42 cm (length) x 32 cm (width) x 60 cm
(height) with a total volume of 80.6 L. The tank has a lid of the same material, and to close
the tank, the lid is tightened with ten screws; ambient air leaks are prevented by a neoprene
O-ring placed between the lid and the tank-body as shown in Figure 1. A waterfall is
generated by commercial 30.5 cm long cascade (DYNASTY SpaParts.com), placed at the
back of the tank. Other cascades were tested (Sect. 3.2). Given that the tank is typically filled
with 40 L of water, the height of the waterfall is about 10 cm from the water. On one side of
the tank, a ½" orifice is used as an air intake. Ambient air passes through a high-efficiency
particulate filter (HEPA - TSI, # 16020551) for particulate matter $\geq 0.3$ µm and a black
carbon filter (PALL, PN 12011) to retain volatile compounds before entering the tank.
Aerosol particles generated in the tank are sampled from the top of the tank through a ¼"
orifice.  A 1.0" orifice at the bottom of the tank is part of the water circulation system.
The circulation system consists of 1.0" internal diameter hoses, PVC pipes, and fittings (Fig.
1), a drain valve at the bottom and an on-off valve. The water is pumped with a centrifugal
pump (Little Giant PondWorks, model 2-MDQ-SC), and the intermittent automatic water


flow is generated and controlled with a corrosion-resistant ½" PVC solenoid valve (WIC
VALVE, model 2PCZ-1/2-D-L) and a programmable time-delay relay (Macromatic Relay,
Model TR 65122). The water flow is continuously monitored with a flowmeter (GPI TM
SERIES, model TM050-N).

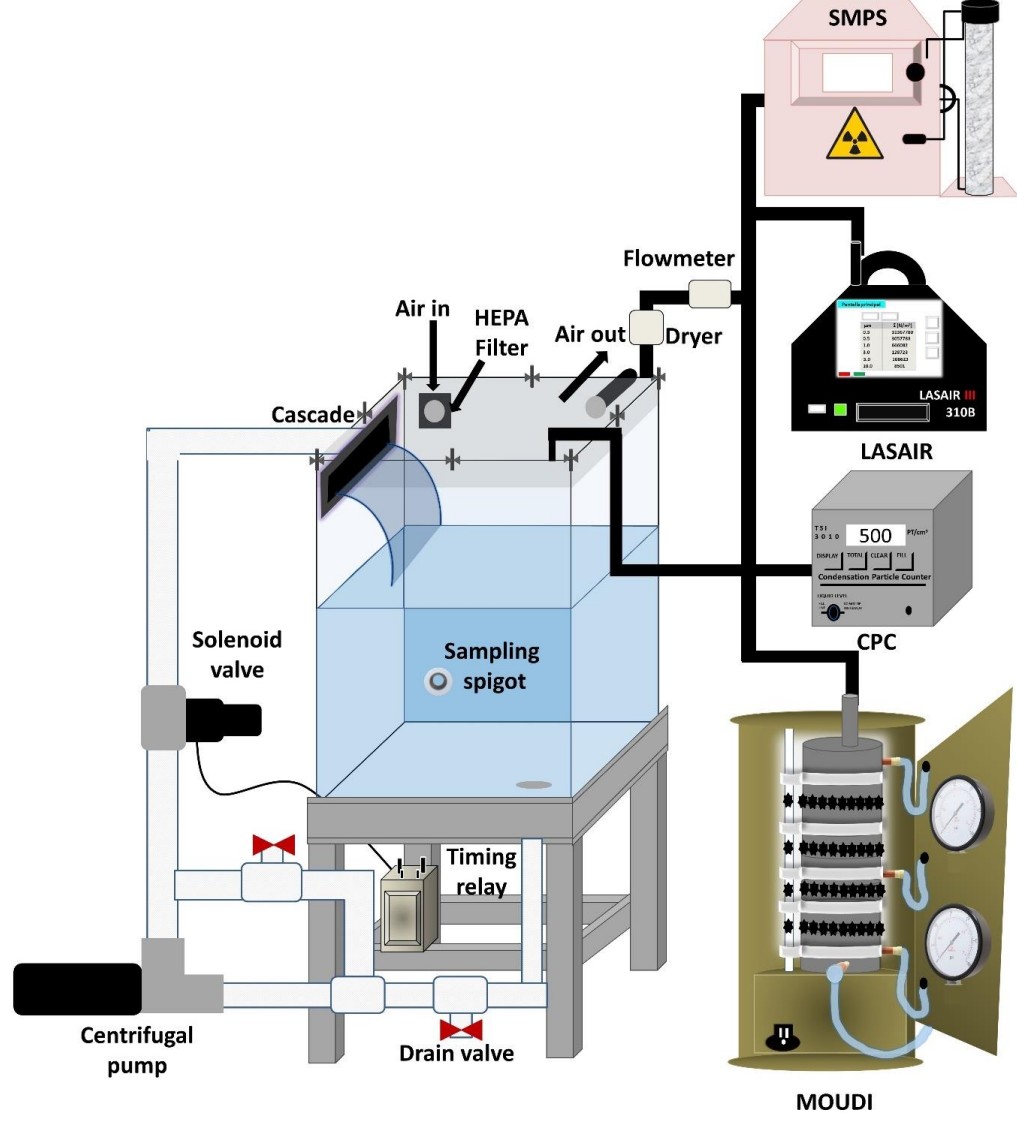


Figure 1. Diagram of the UNAM-MARAT experimental setup and the location of additional
194                                         instrumentation.





Prior to each experiment, the tank is cleaned twice: first with distilled water and then with a
mixture of isopropanol and distilled water. The distilled water and the 10% isopropanol
solution are recirculated for 30 minutes. This procedure is carried out to eliminate residues
and microorganisms from previous experiments. Once the system is cleaned, 10 L of the
sample to be used are added to the tank and recirculated for 30 mins to purge it. Subsequently,
the tank is completely emptied and filled with 40 L of the water sample to be analyzed. The
tank is carefully closed and left to stand overnight. To monitor SSA generation, a
condensation particle counter (CPC 3010, TSI) is connected at the inlet located at the top of
the tank and data is collected for 20 mins to determine the baseline (background)
concentration. Afterward, the waterfall is turned on for 20 minutes to generate aerosol
particles, and samples are then taken for 10 minutes. The waterfall operates intermittently to
mimic wave breaking (the operating time was 2s on and 10s off).
**2.2 Additional instrumentation**
Online and offline measurements were made to characterize the SSA generated in the
UNAM-MARAT. Due to different flow rates of the online and offline instrumentation, not
all instruments sampled simultaneously.
SSA PSD for particles larger than 0.3 µm were measured with an optical particle counter
(LasAir III 310B, Particle Measuring Systems) with cut sizes of 0.3, 0.5, 1.0, 3.0, 5.0, and 10
µm. The data was recorded every 11s and the instrument was operated at a flow rate of 28.3
L min$^{-1}$.
SSA PSD for particles ranging between 10 and 400 nm was measured with a Scanning
Mobility Particle Sizer (SMPS, TSI). The SMPS setup included an electrostatic classifier
(model 3080, TSI), a scanning differential mobility analyzer (DMA, model 3081), and a
water condensation particle counter (WCPC, model 3787). The sample flow rate was set at
0.6 L min$^{-1}$. Measurements were taken in 10 consecutive runs, each lasting 5 minutes, while
the waterfall was in operation.
SSA particles were collected as a function of their aerodynamic diameter using a micro-
orifice uniform deposit impactor (MOUDI 100R, MSP) at a flow rate of 29.9 L min$^{-1}$. The
cut sizes of MOUDI are 0.18, 0.32, 0.56, 1.0, 1.8, 3.2, 5.6 and 10.0 µm (Mason et al., 2015).
Aluminum substrates of 47 mm (TSI) were used for the subsequent chemical composition
analysis, while hydrophobic glass coverslips of 22 mm x 22 mm (HR3-215, Hamptom
Research) were used for the subsequent INP analysis. During a typical experiment, samples
were collected four times for 10 mins each, on the same substrate. The substrates were stored
in sealed petri dishes at 4°C until analyzed.
**2.3 Chemical analysis**



Particles collected on the aluminum substrates were analyzed by ion chromatography. The
substrates were cut and placed inside polyurethane bottles with 10 mL deionized water, and
then placed in an ultrasonic bath (model 3510, Branson) for 1 h at 47°C, allowing for the
desorption and fragmentation of organic and inorganic particles. Subsequently, the bottles
were placed on a mechanical orbital shaker (model 3005, GFL) for 6 h at 350 rpm. Samples
were then filtered with acrodisc syringe filters of 25 mm diameter with a pore size of 0.2 µm
(Pall Corporation). Finally, the filtrate was stored at -4°C (Chow and Watson, 1999). The
identification and quantification of anions ($Cl^-$, $NO_3^-$, $Br^-$, $SO_4^{2-}$, $PO_4^{3-}$) and cations ($Na^+$,
$Mg^{2+}$, $Ca^{2+}$, $NH_4^+$, $K^+$) was performed by a Dionex model ICS-1500 chromatograph equipped
with an electrical conductivity detector. A Thermo Scientific Dionex IonPac AS23-4 µm
Analytical Column (4 mm x 250 mm) with Thermo Scientific Dionex CES 300 Capillary
Electrolytic Suppressor module and the mobile phase was 4.5 mM $Na_2CO_3$ - 0.8 mM NaHCO
at 1 mL min$^{-1}$ flow rate for anions and a Thermo Scientific Dionex IonPac CS 12A Cation-
Exchange Column (4 mm x 250 mm) with the Thermo Scientific Dionex CES 300 Capillary
Electrolytic Suppressor and the mobile phase was a solution of $CH_4SO_3$ 20 mM and 1 mL
min$^{-1}$ flow rate for cations as described in Ladino et al. (2019).
The ice nucleation abilities, via immersion freezing, of the SSA particles were measured
through the droplet freezing technique (DFT). Detailed information on the operation of the
UNAM-DFT can be found in Córdoba et al. (2021); therefore, only a brief description is
provided below. The UNAM-DFT consists of four modules: (i) a cold stage, (ii) a humid/dry
air system, (iii) an optical microscope with a video recording system, and (iv) a data
acquisition system. Each glass coverslip with the SSA is placed on the cold stage and isolated
from the ambient atmosphere. Humid air is circulated through the system, inducing liquid
droplet formation by water vapor condensation. When droplets reach a diameter of 170 µm
(on average), dry air is injected to induce evaporation and to increase the distance between
droplets and, hence, to avoid contact droplet freezing. The humid/dry air system and the
valves of the cold stage are then closed, and the temperature of the sample holder is decreased
from 0 to -40 °C at a cooling rate of 10 °C min$^{-1}$. Droplet freezing is detected when the droplet
changes from bright to opaque as seen during the video analysis. Thus, the freezing
temperature is determined through the data acquisition system.
The ice-active surface site density ($n_s$) was derived from Eq. (1) at -15, -20, -25, and -30°C
following Si et al. (2018):
$$n_s(T) = \frac{[INP(T)]}{S_{tot}} \tag{1}$$

where $[INP(T)]$ is the INP concentration (L$^{-1}$) at temperature (T) and $S_{tot}$ is the total surface
area of all aerosol particles. Full details of the $n_s$ calculation can be found in the Supporting
Information.





The [$INP(T)$] is obtained from Eq. (2) in Mason et al. (2015):

$$[INP(T)] = -ln\left(\frac{N_u(T)}{N_o}\right) \cdot \left(\frac{A_{deposit}}{A_{DFT}V}\right) \cdot N_o \cdot f_{ne} \cdot f_{nu,0.25-0.10\,mm} \cdot f_{nu,1\,mm} \quad (2)$$

where $N_u(T)$ is the number of unfrozen droplets at a temperature $T(°C)$, $N_o$ is the total number of droplets (dimensionless), $A_{deposit}$ is the total area of the aerosol particles deposited on the MOUDI hydrophobic glass coverslip (cm$^2$), $A_{DFT}$ is the area of the sample analyzed by the DFT (cm$^2$), $V$ is the volume of air through the MOUDI (L), $f_{ne}$ is a correction factor to account for the uncertainty associated with the number of nucleation events in each experiment (dimensionless), and $f_{nu}$ is a correction factor to account for changes in particle concentration across each MOUDI sample (dimensionless).

**Collection of ocean water samples**

The seawater samples to generate the SSA with the UNAM-MARAT were collected at three different Mexican coastal sites (Figure 2): The Port of Veracruz (PoV, Veracruz), the Bay of Acapulco (BoA, Guerrero), and the Bay of Santiago-Manzanillo (BoSM, Colima). The coordinates are given in Table S1. Approximately 60 L of seawater were collected in 20 L polyethylene containers previously washed with distilled water and purged with seawater. The samples were transported to Mexico City at room temperature. In the case of the BoSM samples, the UNAM-MARAT was deployed to the Water Quality Laboratory located in the University Center for Oceanological Research, University of Colima, Manzanillo, where some experiments were carried out in-situ. A second 60 L of seawater from the BoSM sample was collected (04/09/2022) and transported to Mexico City to evaluate potential changes that may occur during transportation. Before introducing the seawater into the UNAM-MARAT, the samples were filtered with a 50 µm mesh to remove some debris and zooplankton.





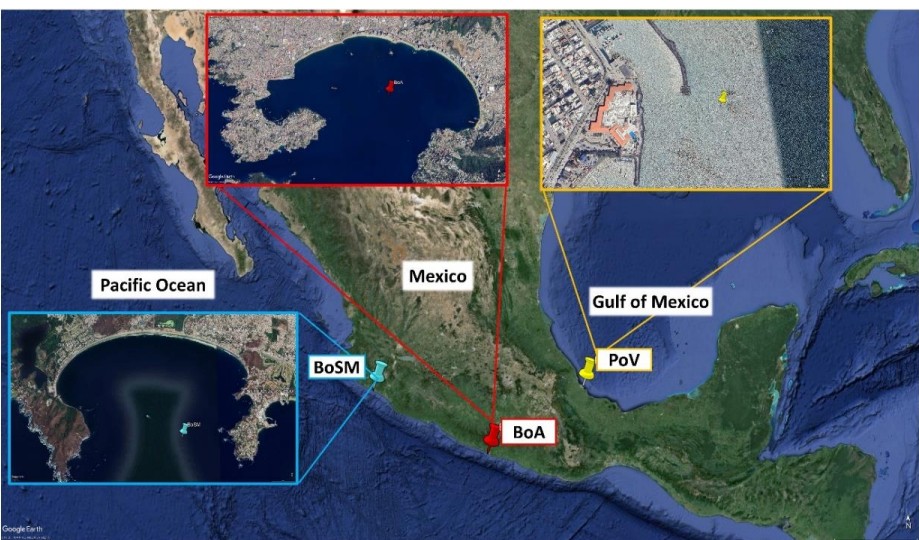

Figure 2. Map showing the sampling locations: The Bay of Acapulco (BoA, red icon), the Port of Veracruz (PoV, yellow icon), and the Bay of Santiago-Manzanillo (BoSM, blue icon). Photo from © Google Earth.

## 3 UNAM-MARAT validation

### 3.1 Background particle concentrations

Air entering the UNAM-MARAT was filtered to ensure that the measured aerosol particles corresponded solely to those generated by seawater and not due to leaks in the tank. Note that filtered air passing through the CPC resulted in an aerosol concentration of less than 0.1 $cm^{-3}$.

Background experiments measured total particle concentrations with the CPC when the tank was filled only with commercial distilled water. An individual experiment consists in measuring the particle concentration for 20 min. This procedure was repeated over three consecutive days, both in the morning and the afternoon (local time). In total, 15 runs were conducted, and the results were averaged with their corresponding standard deviation. Figure 3 shows the average particle concentration from the tank when it was filled with distilled water with the cascade off (black line) and with the cascade on (blue line). The shaded areas represent the standard deviation of each curve. The average particle concentration oscillated between 7.8 and $13.2 \pm 2.3$ $cm^{-3}$ with the cascade off (the top left figure shows a zoom of the base line), which indicates that there is a low number of particles within the tank. These results are in accordance with those reported by Prather et al. (2013), who found a baseline $< 20$ $cm^{-3}$ in the Wave Channel. When the cascade was in operation, aerosol particles were





generated from the distilled water (up to 313 cm$^{-3}$), indicating that the water used was not
completely free of particles. Also, given that the samples were not passed through a diffusion
dryer, it is likely that the measured particles correspond to large water droplets that did not
evaporate before entering the CPC.
A commercial sea salt (i.e., *Instant Ocean Sea Salt, IOSS*) was used as a proxy for sea water
for the validation of UNAM-MARAT. For a typical experiment a solution was prepared in
distilled water, achieving a salinity of $28.8 \pm 0.2$ g L$^{-1}$. The tank was filled with 40 L of an
IOSS's solution and the total particle concentration was measured with the CPC. The average
particle concentration is represented by the red line in Figure 3. It shows that the maximum
concentration observed after 20 min of turning on the cascade was 1016 cm$^{-3}$. Figure 3
demonstrates that the UNAM-MARAT is capable of generating SSA, as indicated by the
observed increasing concentration.

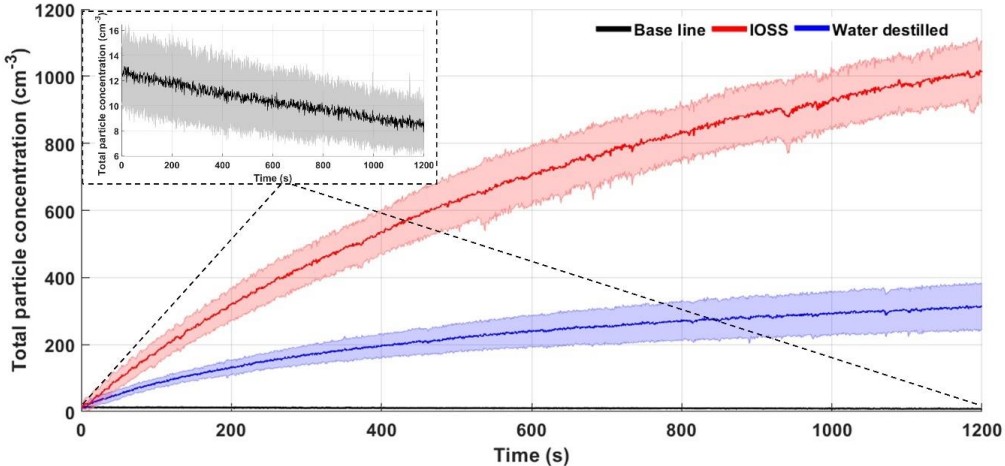


Figure 3. Total aerosol concentration as a function of the time (s) with the cascade off (black line in
the inset in the upper panel), with the cascade on with distilled water (blue line) and with an IOSS's
solution (red line). The shaded areas represent the standard deviation of each curve.

### 3.2 Cascade test

To evaluate the role that the cascade plays in SSA generation in the UNAM-MARAT, the
tank was filled with 40 L of an IOSS's solution and the total particle concentration was
measured when using four different cascades: one was a commercial cascade and the other
three were homemade. The homemade waterfalls consisted of cylindrical PVC pipes
featuring a slot designed to facilitate the formation of a plunging water sheet. An internal
tube with multiple evenly spaced holes was incorporated to enhance water distribution as
shown in Figure S1. The main characteristics of each cascade produce with varying slot
lengths, inner tube diameters and number of holes, are shown in Table S3.





The cascade A produced the lowest particle number concentrations, whereas the highest concentrations were observed with cascades C and D (Fig. 4). As shown in Table S3, the slot length of cascade D is longer than the other cascades, suggesting that the slot's length is a key factor in increasing particle generation.

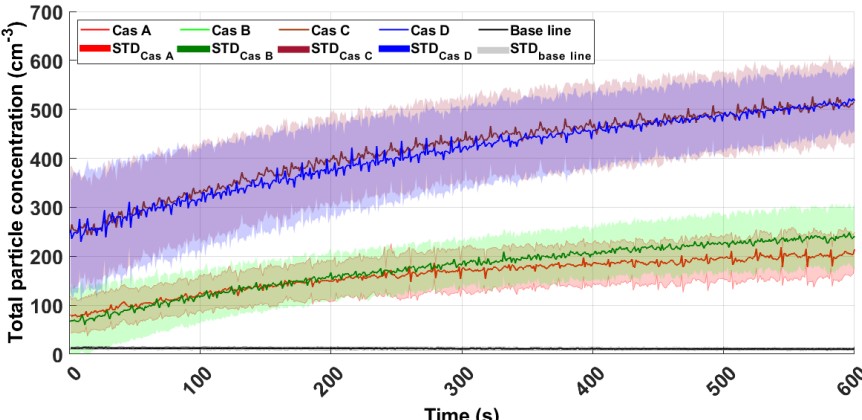

Figure 4. Total aerosol number concentration as a function of the time (s) for the different cascades. Each curve represents the average of fourteen experiments, with the shaded area representing the corresponding standard deviation.

In contrast, experimental results suggest that the number of holes in the inner cascade tube, and its diameter, play a secondary role in particle production. A longer slot length results in a larger artificially generated wave, through the plunging sheet of water mechanism rather than the plunging jet mechanism.

Stokes et al. (2013) noted that the shape and penetration of water drops (i.e., plunging sheet or plunging jet) affect aerosol particle production, with plunging sheet generating more particles. In contrast, other aerosol generation systems used in marine tanks have shown limited efficiency, as they often produce particles within a narrow size range. For instance, Fuentes et al. (2010) observed that systems employing glass frits and aquarium diffusers can produce high concentrations of particles, but these particle size typically range between 0.012 and 0.018 µm. This limitation arises because these systems primarily simulate the film drop mechanism. Plunging sheet systems, such as in the UNAM-MARAT, can produce a broader range of particles sizes, as they facilitate both the film and jet drop mechanisms, leading to more diverse aerosol size distributions (Stokes et al., 2013)

### *3.3 The intermittency time*

Waves in marine environments are not generated continuously, mainly due to the different energy processes that drive them, their intensity, and physical and physiographic aspects, resulting in an intermittent behavior (Jelley, 1989). Wang et al. (2017) revealed that a





continuous cascade complicates the rupture of bubbles on the water surface, similarly
affecting the formation of aerosol particles through the jet drops mechanisms, which is
important in generating supermicron particles. For this reason, the solenoid valve controls
the intermittence of the cascade and the operating time is modulated with a time-delay relay.
Five intermittency values were evaluated: 2 s on / 10 s off, 4 s on / 10 s off, 2 s on / 4 s off,
2 s on / 6 s off, and 2 s on / 8 s off. The highest particle concentrations were observed for the
2 s on / 4 s off (~2400 cm$^{-3}$, Figure 5c) and 2 s on / 6 s off (~2000 cm$^{-3}$, Figure 5d)
configurations, followed by the 2 s on / 8 s off (~1600 cm$^{-3}$, Figure 5e). For the 2 s on / 10 s
off (Figure 5a), 4 s on / 10 s off (Figure 5b) configurations, the maximum aerosol
concentrations were very similar (about 1400 cm$^{-3}$ after 600 s). Harb and Foroutan, (2019)
also evaluated the role of the intermittency (i.e., 3 s on and 1s off, 3 s on and 2 s off, 3 s on
and 3 s off, 3 s on and 4 s off, and 3 s on and 5 s off). The authors conclude that using a
longer pause time to allow the bubble plume to develop, is beneficial for facilitating the
mechanisms of film and jet drops production. Although the 2 s on / 10 s off configuration did
not report the highest particle concentration in the UNAM-MARAT, in the remainder
experiments we choose this configuration to be comparable to the configuration used in
Stokes et al. (2013). Additionally, using configurations 2 s on / 4 s off and 2 s on / 6 s off
tend to create a more continuous plunging sheet, which could affect the size of the aerosol
particles produced and may not accurately simulate the natural wave breaking processes.

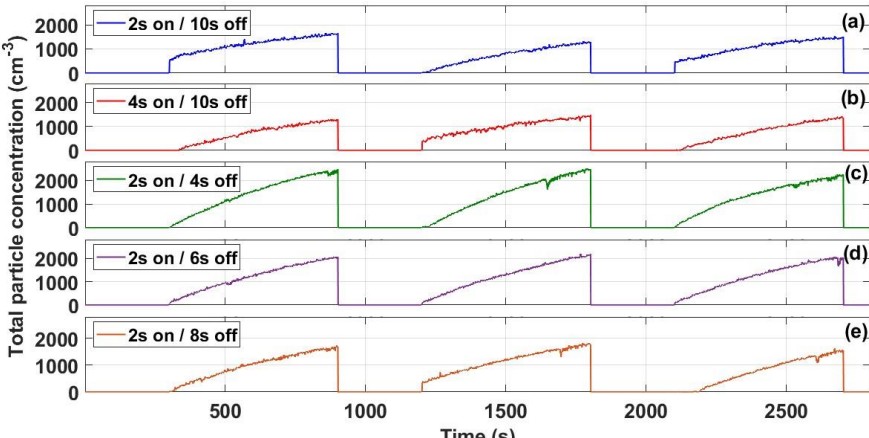


Figure 5. Total aerosol number concentration as a function of the time (s). The time series represent
experiments with different intermittence values evaluated from an IOSS's solution in the UNAM-
MARAT. The different panels correspond to (a) 2 s on 10 s off, (b) 4 s on 10 s off, (c) 2 s on 4 s off
385                        (d), 2 s on 6 s off, and (e) 2 s on 8 s off.

Bates et al. (1998) reported that SSA concentrations measured in natural marine
environments, such as the Southern Ocean were < 500 cm$^{-3}$. Concentrations achieved with
the Wave Channel vary between 50 and 100 particles cm$^{-3}$, while experiments using artificial





seawater (i.e., a salt mixture) conducted with the MART have reported particle
concentrations ranging from 680 to 1053 cm$^{-3}$ (Thornton et al., 2023). The differences in the
concentrations of particles generated in the MART, the Wave Channel, and the UNAM-
MARAT tanks can be attributed to several factors, including the aerosol generation
mechanism, the composition of the used seawater, and the specific design of each tank. For
instance, the Wave Channel uses a paddle to create a disturbance for a wave generation,
which affects aerosol production. In contrast, although the MART and the UNAM-MARAT
employ similar mechanisms for aerosol generation, the particle concentration differences
may be due to the different tank sizes: 210 L (MART) versus 80 L (UNAM-MARAT).
*3.4 Waterfall height*
The importance of the waterfall height was assessed by testing different volumes of an
IOSS's solution (salinity 28.8 ± 0.2 ppt). The total aerosol number concentration was
measured for the following water volumes: 20, 30, 40, and 50 L which resulted in a waterfall
height of 38.5, 30.5, 22.5, and 14.5 cm, respectively. Figure 6a shows the average total
particle concentration (blue line) with their corresponding uncertainty (shaded area). The
experiments for each water volume were performed over three different days.

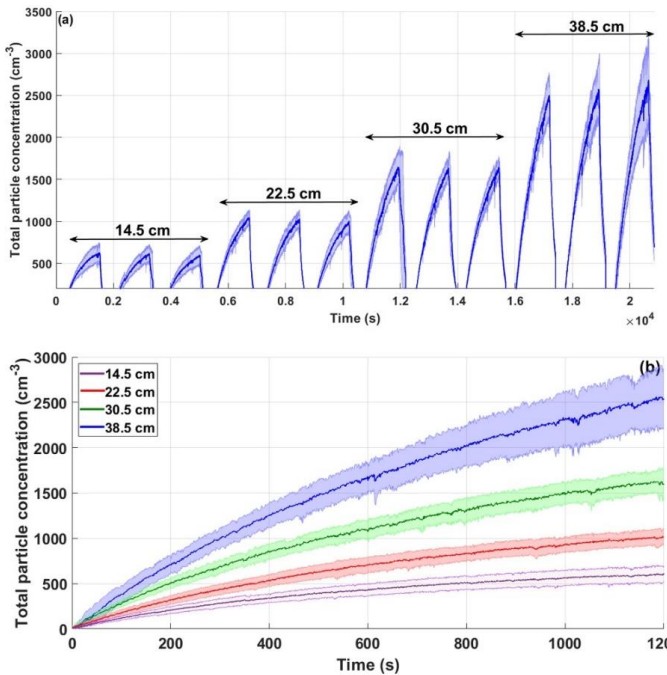


Figure 6. Total aerosol concentration as a function of the time (s). (a) the blue line shows the
average of three days of repetitions from an IOSS's solution in the UNAM-MARAT. The shaded
area represents the uncertainty of those repetitions. (b) Comparison of the average aerosol particle
concentrations generated from different volumes with their corresponding uncertainty.





The highest concentrations were observed for the largest waterfall height (38.5 cm, 20L of water), with concentrations up to 2500 cm$^{-3}$, followed by the waterfall height 30.5 cm (30 L of water), which reported a maximum concentration of 1600 cm$^{-3}$. The lowest concentration (600 cm$^{-3}$) were recorded for the waterfall height 14.5 cm (50 L of water). Notably, the uncertainty was low at the beginning of the experiments but increased over time (Figure 6b). Although the concentrations for the waterfall height 22.5 cm (40 L of water) were not as high as those reported for the waterfall height 38.5 cm, the uncertainty remained lower throughout the aerosol particle emission process. Therefore, this height was selected for most of the following experiments.

### 3.5 Particle size distribution

The final step in characterizing the UNAM-MARAT was to evaluate the PSD of the generated SSA. The particle monitoring was conducted using a SMPS and a LasAir, to assess if the UNAM-MARAT could generate particles across a wide size range. The tank was filled with 40 L of an IOSS's solution and ten experiments were carried out using the intermittent cascade (2 s on, 10 s off). Figure 7a shows the PSD obtained with the SMPS for particles ranging between 10 nm and 400 nm. The black line represents the average of the ten experiments, and the area between the blue lines indicates the standard deviation. A peak in concentration for particles between 0.1 and 0.2 µm in diameter was observed, corresponding to the accumulation mode. This mode is consistent with data reported using the MART (Stokes et al., 2013). The PSD obtained with the LasAir for particles ranging between 300 nm and 10 µm is presented in Figure 7b. The highest concentration was observed in the size bin corresponding to the smallest particles (i.e., 0.3 – 0.5 µm). These results demonstrate that the UNAM-MARAT can generate marine aerosol particles with sizes ranging from 30 nm to 10 µm.

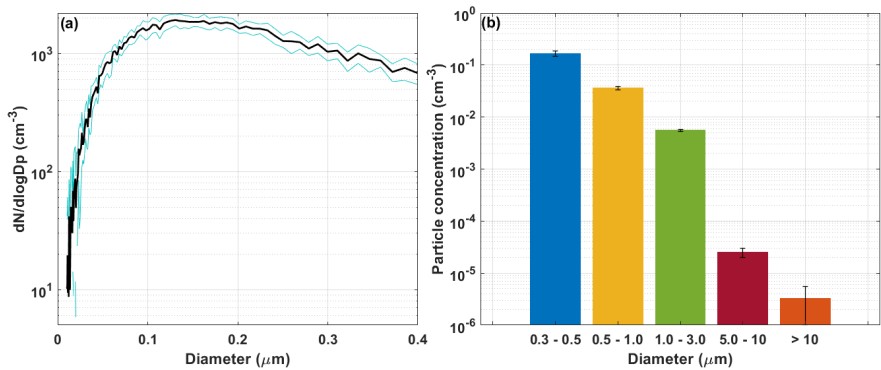

Figure 7. Aerosol particle size distribution obtained with the IOSS solution using (a) the SMPS and (b) the LasAir.



## 4 Case Study: Ice nucleating abilities of SSA

### 4.1 Aerosol particle concentrations and PSD

This section presents the results obtained after generating SSA in the UNAM-MARAT using water samples from BoA, PoV, and BoSM. The results from BoSM correspond to samples collected on April 9, 2022, which were transported to Mexico City, as was the case with the BoA and PoV samples. The SSA generation experiments were performed on April 25, 2022. The highest number of particles generated with the UNAM-MARAT were obtained from the BoSM samples, showing concentrations up to 2000 cm$^{-3}$. In contrast, the lowest concentrations were observed from the PoV samples (570 and 590 cm$^{-3}$). This variation related to the origin of the samples may be due to differences in composition (inorganic and organic matter), as the equipment used to generate the SSA was the same and the protocol followed was identical. It is worth noting that at the time of collecting the BoSM samples, the water appeared very turbid, likely due to the decomposition of organic matter. In comparison, the BoA and PoV samples had a clearer appearance.

Mayer et al. (2020) reported particle concentrations ranging from 400 to 500 cm$^{-3}$ in an experiment with seawater collected at Scripps Pier, to which nutrients were added to promote phytoplankton blooms. Thornton et al. (2023) emphasize the importance of seawater composition in particle concentration. The authors conducted experiments in the MART, creating mesocosms with the *Thalassiosira weissflogii* diatom and the *Synechococcus elongatus* cyanobacterium, observing particle concentrations ranging from $1 \times 10^6$ to $2 \times 10^6$ cm$^{-3}$ for both species, with peaks reaching up to $6 \times 10^6$ cm$^{-3}$.

The PSD (for particles larger than 300 nm) from the different samples were comparable as shown in Figure 8. The highest concentrations were observed for particles in the smallest size bin i.e., 0.3 and 0.5 µm. Out of the three samples, the highest concentrations were observed in the BoSM samples for particles with diameters between 0.5 and 10 µm and in the PoV samples for particles between 0.3 and 0.5 µm. The numbers at the top of the bars in Figure 8 correspond to the average concentrations. Figure 8 also shows that the UNAM-MARAT can produce coarse-mode particles (> 1 µm). Generally, particles in this size correspond to sea salt (NaCl) and biological particles (intact or fragmented cells of bacteria, phytoplankton, macrogels, and transparent exopolymer particles, TEP) (Prather et al., 2013; Verdugo et al., 2004). Given that the most efficient INPs are likely particles > 500 nm (DeMott et al. 2010), the PSD for the ambient samples was only monitored using the LasAir (particles > 300 nm).

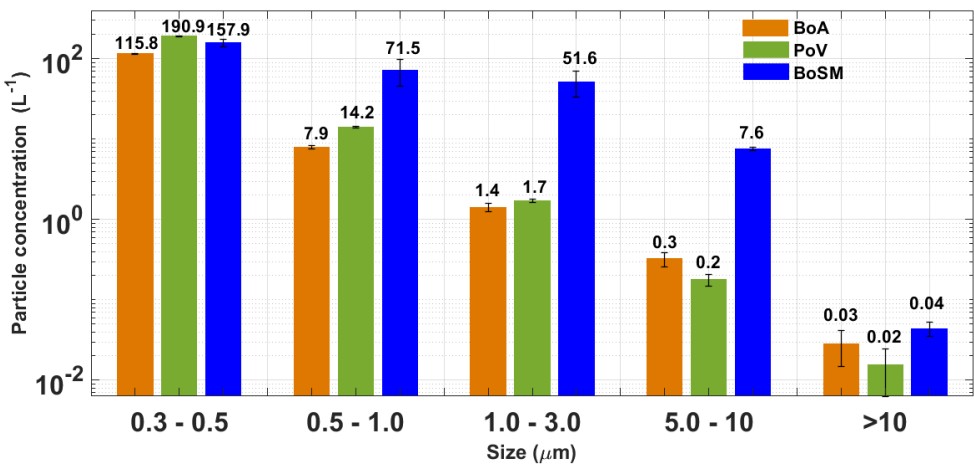

Figure 8. Aerosol particle size distribution for BoA (orange bars), PoV (green bars), and BoSM (blue bars). The error bars represent the standard deviation for each size bin.

### 4.2 Chemical composition

To understand the differences in the chemical composition between samples, the concentration of ions as a function of particle size was analyzed for the PoV and BoSM samples (Figure 9). The BoA sample could not be processed for this specific analysis due to unintentional technical issues. As expected, the dominant ions were $Na^+$ and $Cl^-$ in both samples. Their concentration was highest for the largest particles (5.6 to 10 μm) and it was lowest for the smallest sizes (0.32- 0.56 μm). An opposite trend was observed for $Ca^{2+}$ and $Mg^{2+}$, as their concentration increases with the particle size. Generally, $Ca^{2+}$ and $Mg^{2+}$ can interact with organic compounds such as carbohydrates, proteins, and lipids. Chin et al. (1998) demonstrated that a proportion of exopolymers present in seawater can assemble into gels through the chelation of $Ca^{2+}$ and $Mg^{2+}$, which form bridges between adjacent or different dissolved organic carbon (DOC) chains.



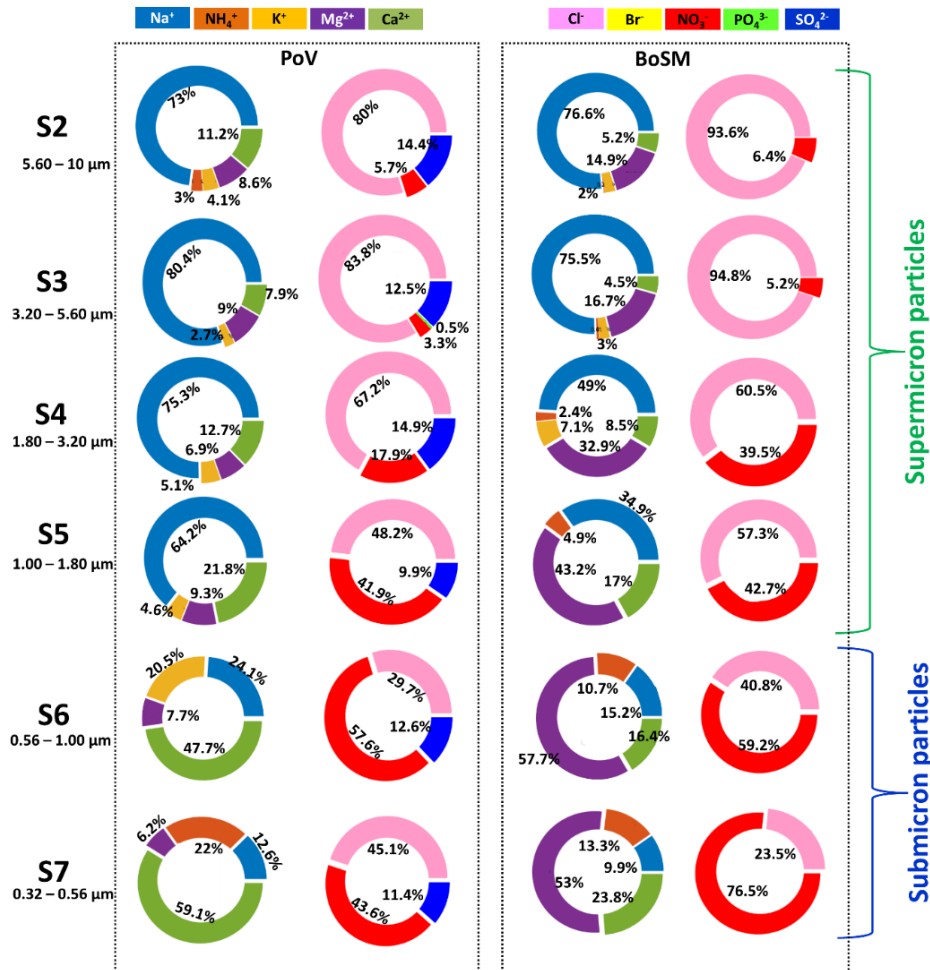

Figure 9. Ion concentration (mg L$^{-1}$) for each MOUDI stage for water samples collected in PoV (left panel), and BoSM (right panel). The pie charts on the left of each group represent cations, while those on the right represent anions.

Regarding the other ions, $NO_3^-$ showed a similar behavior as $Ca^{2+}$ and $Mg^{2+}$, while $NH_4^+$ increases with size from stage 4 (particle size: 1.80 - 3.20 µm) to stage 7 (particle size: 0.32 – 0.56 µm) for the BoSM sample. The presence of ions such as $NO_3^-$ and $NH_4^+$ may be due to the decomposition of organic matter or excretions of phytoplankton and zooplankton as well as the availability of nutrients in the environment of terrigenous origin by runoff or continental wind input (Anderson et al., 2002). The presence of $SiO_4^{-2}$ in the PoV sample may be the result of the dissolution of minerals rich in silicates of continental origin or of the dissolution of ortho silicic acid ($H_4SiO_4$) that is used in the biogeochemical cycle that is also





regulated by phytoplanktonic and zooplankton organisms (Kuuppo et al., 1998; Wu and
Chou, 2003).
Bigg and Leck (2008) showed that particles with diameters smaller than 200 nm were
exopolymers produced by bacteria and algae, as well as microgels formed from these
exopolymers in laboratory experiments. Furthermore, the chemical composition of these
particles is closely linked to biological activity. For instance, Facchini et al. (2008)
demonstrated that submicron particles collected in the eastern North Atlantic off the coast of
Ireland were predominantly composed of organic constituents. Russell et al. (2010) found
that the majority of the organic components in submicron aerosol particles collected in the
Arctic consisted of organic hydroxyl groups (including polyols and alcohols) characteristic
of saccharides. Similarly, Bates et al. (2012) suggested that the organic mass from aerosol
particles collected off the coast of California was composed of carbohydrate-like compounds
containing organic hydroxyl groups, alkanes, and amines. Our results demonstrate that
supermicron particles are largely dominated by sodium chloride. Additionally, our findings
indirectly suggest that submicron particles also contain significant amounts of organic
material, consistent with the findings reported by Prather et al. (2013).

### 514 *4.3 INPs*

Figure 10a shows the concentration of INPs for the three set of samples as a function of
temperature. The temperatures at which the different samples were able to nucleate ice, via
immersion freezing, were found to be -19 to -34°C, -18 to -34°C, and -18 to -33°C for the
BoA, PoV, and BoSM samples, respectively. The measured INP concentration ranged from
0.9 to 95.4 L$^{-1}$ for BoA, 1.7 to 97.5 L$^{-1}$ for PoV, and 0.9 to 130.7 L$^{-1}$ for BoSM. From these
results, it can be inferred that there are no significant differences in the INPs concentrations
among the water samples collected from the three sites.

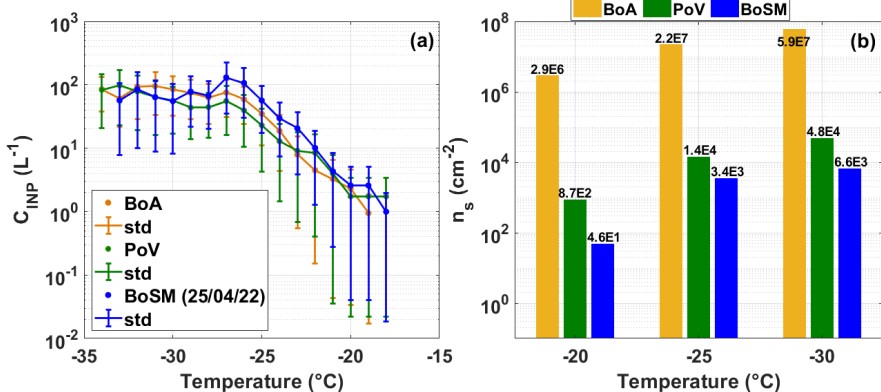

Figure 10. Comparison of the ice nucleating abilities between samples. (a) INP concentration as a
function of temperature and (b) $n_s$ values as a function of temperature. Yellow, green and blue
correspond to the BoA, PoV, and BoSM samples, respectively.



In an experiment conducted using the MART with seawater collected near SIO, DeMott et
al. (2016) found that the INP concentrations varied between $1 \times 10^{-3}$ and $1 \times 10^3 \, L^{-1}$, with ice
nucleation temperatures ranging from -7 to -30°C. The results observed in the marine aerosol
samples generated with the UNAM-MARAT (this study) fall within the range reported by
DeMott et al. (2016). Additionally, the findings in this study are consistent with those
reported by McCluskey et al. (2017), who found that particles generated in the MART with
waters collected near SIO, stimulated to produce phytoplankton blooms, were able to
nucleate ice between -7 and -32°C, with INP concentrations ranging from $1 \times 10^{-3}$ and $1 \times$
$10^1 \, L^{-1}$. On the other hand, Thornton et al. (2023) reported that aerosol particles generated
using the MART from waters containing *Thalassiosira weissflogii Synechococcus elongatus,*
as previously mentioned, exhibited ice freezing temperatures between -14 and -32°C. DeMott
et al. (2016) and Thornton et al. (2023) concluded that the warmer freezing temperatures
observed in their experiments coincided with peaks in chlorophyll a (Chl-a) in their
mesocosms. In contrast, McCluskey et al. (2017) demonstrated that increases in INPs active
between -25 and -15°C lagged behind the peak in Chl-a, suggesting a consistent population
of INPs associated with the collapse of phytoplankton blooms. The difference with the
experiments conducted using the UNAM-MARAT is that no culture medium was added to
induce blooms in our experiments.
As mentioned earlier, $n_s$ is a robust and quantitative metric for comparing the ice nucleating
abilities of aerosol particles (Holden et al., 2021). Therefore, $n_s$ was calculated for each
sample, as shown in Figure 10b. It was found that the highest and lowest $n_s$ values were
derived from the BoA and BoSM samples, respectively. Although the BoSM sample had the
highest particle concentration among the three analyzed samples (Figure 8), it exhibited the
lowest $n_s$ values, indicating that the particles emitted from this water sample have fewer
active sites for ice nucleation. DeMott et al. (2016) and McCluskey et al. (2017) report high
$n_s$ values on the order of $1 \times 10^5$ y $1 \times 10^6 \, cm^{-2}$, respectively, which are consistent with the BoA
values found in this study (Table S4).

### 4.4 Correlation of $T_{50}$ with Ion Concentration

Spearman's correlation coefficients were calculated between the concentration of certain ions
(since some could not be determined in specific particles sizes) and the median freezing
temperature ($T_{50}$) to evaluate if the ice nucleation efficiency is associated with organic matter
(Figure S2). A better correlation was observed in the BoSM samples than the PoV sample.
Moreover, the highest Spearman coefficients were found for the samples collected on the
second day at BoSM, when the waters were turbid. Considering that the high concentrations
of $Ca^{2+}$ and $Mg^{2+}$ ions are associated with continental particles that promote primary
productivity in the coastal zone, their subsequent remineralization could mean that the BoSM
sample collected on 04/09/22 was enriched in organic material, which explains the high ice
nucleation efficiency observed in this sample.



### 4.5 Analysis of transport in the Manzanillo seawaters samples.

Additional experiments were carried out with the BoSM samples to evaluate whether the transport of samples from the sampling site to our laboratory located in Mexico City affects the ice nucleating abilities of SSA generated in the UNAM-MARAT. Two water samples were taken in the BoSM. The 09/04/22 BoMS "fresh sample" results refer to experiments conducted on the second day after collection (experiments conducted in the field), and the 25/04/22 BoMS "aged sample" results refer to the experiments conducted 15 days after collection (experiments conducted in Mexico City). The sample was not preserved to maintain conditions similar to those applied to the BoA and PoV samples.

The particle concentration was higher in the aged sample for the sizes between 0.3 and 1.0 µm and those >10 µm. However, for particles between 1.0 and 10.0 µm, the highest concentrations were observed on the fresh samples as shown in Figure S3a.

Since it was observed that aging impacted the number of particles, the impact of aging on the ice nucleating abilities was also analyzed. The $n_s$ values were calculated for three temperatures (i.e., -20, -25, and -30°C). Figure S3b shows that the $n_s$ values were consistently higher in the aged sample. This could indicate that biological activity continued during the transport of the samples, which might explain the increase in $n_s$ values. However, ion concentrations did not change significantly between the fresh and aged samples. It is advisable to perform other chemical analyses to validate this hypothesis.

## 5 Conclusions

The UNAM-MARAT was specifically designed to simulate waves breaking to generate sea spray aerosol and to evaluate the ability of marine aerosol particles to act as INP. The ideal conditions established to work with the UNAM-MARAT were to use 40 L of seawater in the tank, employ a cascade with a slot length of 28.3 cm, and a plunging sheet intermittent cascade with an operating configuration of 2 s on and 10 s off to achieve particle concentrations exceeding 1000 cm$^{-3}$.

The UNAM-MARAT has proven to be an effective tool for evaluating and analyzing the physical and chemical properties of SSA from seawater samples collected at various locations. It offers a cost - effective alternative to expensive field campaigns, providing a controlled and reproducible method for simulating natural SSA generation. The results obtained from the UNAM-MARAT during its characterization are comparable to those obtained from other wave tanks, confirming its reliability and suitability for marine aerosol studies.

From the case study, we were able to successfully generate marine aerosol in the laboratory using seawater samples from various coastal areas of Mexico. The aerosol reached particle concentrations up to 2000 cm$^{-3}$ across a wide range of particles sizes, from 10 nm to 10 µm.



Additionally, our results show that the Mexican oceanic waters contain INP with concentrations up to 130.7 $L^{-1}$. Among the three seawater samples analyzed, the BoA sample exhibited the highest ice nucleation abilities, based on the ice active site density values measured between -20 and -30°C. Furthermore, our findings reveal a direct relationship between particle size and composition. Larger particles (>1 µm) were found to be enriched in NaCl, whereas smaller particles showed an increased fraction of $Ca^{2+}$, $Mg^{2+}$, and $NO_3^-$ related to the presence and degradation of organic matter.

The development of the UNAM-MARAT device and the comprehensive analysis of aerosol particles from different coastal regions contribute significantly to our understanding of the role of marine aerosol particles in mixed cloud formation and in the regional precipitation patterns. The newly built tank will serve as a valuable tool for future atmospheric and environmental studies.

*Data availability*. Data are available upon request to the corresponding author.

*Author contributions*. MFC, RC, and LAL designed and built the thank. MFC, GBR, and LAL designed the field campaign and the experiments. MFC, AO, GC, IM, and LAL carried out the field measurements and the collection of ambient samples. MFC, DRR, HAO, and TC performed the chemical analysis. MFC and LAL wrote the paper, with contributions from all coauthors.

*Competing interest.* The authors declare that they have no conflict of interest.

*Acknowledgments.* We thank Gabriel García, Manuel García, Omar López, and Victor García for their invaluable assistance in the construction and maintenance of the UNAM-MARAT. Also, we would like to thank Kenia Villela, Daniela Leal, María Isabel Saavedra, Dalia Aguilar, Eva Salinas, and Leticia Martínez for their assistance with the experiments related to the UNAM-MARAT. This study was financially supported by the PAPIIT IN111120 grant, the CONACYT fellowship for doctoral students, and the Marcos Moshinsky Foundation.

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
