# Peer review of "The UNAM-MARine Aerosol Tank (UNAM-MARAT): An Evaluation of the 1 Ice-Nucleating Abilities of seawater from the Gulf of Mexico and the Mexican 2 Pacific 3 M. Fernanda Córdoba1, Rachel Chang2, Harry Alvarez-Ospina3, Aramis Olivos-Ortiz4 4 Graciela B. R"

_Atmospheric Measurement Techniques, 2024_

## Author Comment (AC1)

We would like to thank both reviewers for their constructive suggestions, which have significantly contributed to improve the quality and clarity of our manuscript. We carefully considered each comment and have made the necessary modifications accordingly. Specific responses to the reviewers' comments, along with the corresponding changes in the manuscript, are provided below in red text.

**Reviewer 1**

General comments:

This paper is written on the development, characterization, and application of a marine aerosol generator. The authors provide a very thorough review of relevant literature both in the introduction and throughout the manuscript. Multiple tests were conducted to validate the results of their system, while also comparing to results of similar systems. The system is then applied to several samples to compare their size distributions, chemistry, and INP concentrations. The results are thorough and make for a linear story as well.

A/ We sincerely appreciate the reviewer's thorough and detailed assessment, as well as her/his positive feedback on the manuscript.

Specific comments:

• Lines 164-166: Cite or mention Stokes et al. as reference for system

A/ Thank you for the suggestion. The following sentence was added to the revised manuscript:

**Lines 167**: "similar to the Stokes et al. (2013) tank"

• Line 178: Provide brief explanation on why the chosen cascade was the most suitable.

A/ We thank the reviewer for the suggestion. The following text was added to the revised manuscript:

**Lines 180-181**: "Other cascades were tested; however, the commercial cascade was selected as it generated the highest concentration of aerosol particles"

• Are OPC counts biased towards the lowest bins?

A/ The used OPC is able to measure aerosol particles between 0.3 and 10 μm. As shown below in Figure A1 from Prather et al. (2013), the particle size concentration around 0.3 μm is always higher than larger particles. Based on the results from Prather et al. (2013) and other studies focusing on laboratory and ambient marine aerosols, we are convinced that the behavior of our OPC is correct, i.e., there is not a bias towards lower sizes.

[Figure]

*Figure A1. Probability density function of the resulting SSA number distributions (dN/dlogdp, with the dp at 15 ± 10% RH) produced by three different methods (for breaking waves (gray squares), plunging waterfall (blue circles), and sintered glass filters (red line)). Modified from Prather et al. (2013).*

- 3.1: Which cascade was used for the background particle test?

  A/ The results shown in Figure 3 were obtained using the commercial cascade. The following sentence was added to the revised manuscript for clarity:

  **Line 311**: "water with the commercial cascade off (black line) and with the commercial cascade on (blue line)"

- 3.1: At what (time) point does the SSA reach a steady state? This information would be useful.

  A/ The steady state was typically observed after 20 minutes. To acknowledge this, the following text was added to the revised manuscript:

  **Lines 209-211:** "The samples were collected after 20 minutes of aerosol generation given that this time was set as the point where SSA reached a steady state."

- 3.2: Which cascade is the best/most realistic? Should be explicitly stated here.

  A/ Thank you for the observation. As seen in Section 3.2, cascades C and D showed the highest concentration. However, cascade D was chosen for the subsequent experiments because its standard deviation was slightly lower than that of cascade C. Additionally, Cascade D was easier to clean which was helpful to avoid interferences in later experiments. Cascade C has inner holes, which might have made the cleaning

procedure more difficult and uncertain when working with samples collected from the ocean.

The following sentence was added to the revised manuscript:

**Lines 345-347**: "Cascade D (the commercial one) was selected for the subsequent experiments because it produced the highest particle concentration, its standard deviation was slightly lower than cascade C and it was the easiest to clean".

- It should be mentioned how the longer delay also allows for resettling/reformation of the SML, which is important for the chemistry of the particles.

    A/ We thank the reviewer for the suggestion. The following text was added to the revised manuscript to provide a better discussion.

    **Lines 383-385**: "However, it is important to note that the longer delay also allows for reformation of the SML, which is important in the composition of the marine aerosol"

- Line 179 states that with 40 L of water, the waterfall is roughly 10 cm, however Lines 401-402 state that a 40 L fill results in a 22.5 cm waterfall height. This is a 2x difference. Please address this discrepancy.

    A/ Thank you for detecting this unintentional mistake. The correct height is 22.5 cm, therefore, this value corrected in **Line 182** in the revised manuscript.

- Line 464-465: Not necessary to state here that coarse mode particles can be produced. This point is made and should be stated in section 3.5

    A/ We agree with the reviewer. Therefore, the sentence was moved to Section 3.5 in the revised manuscript.

    **Lines 439-440**: "Additionally, it is important to highlight that the UNAM-MARAT can produce coarse-mode particles (> 1 µm)."

    The information in the Section 4.1 was slightly modified.

    **Line 486**: "Generally, the coarse particles correspond…"

- The authors reference how similar works induced phytoplankton blooms and that is what accounts for varying INP concentrations in other works. A comparison of INP concentrations from their work to INP concentrations from the start of the blooms (prior to culture additions) for other works would be a more appropriate comparison and useful addition.

A/ Thank you for this valuable suggestion. We acknowledge the importance of comparing INP concentrations from our work to those measured at the start of phytoplankton blooms in similar studies. A brief discussion was added as follows.

**Lines 564-570:** "When comparing our results with the abovementioned studies (i.e., McCluskey et al. 2017; DeMott et al. 2016) before the addition of the culture medium (Day 0), we find that our INP concentrations are rather comparable with the values reported by both studies. However, the clear difference between the former studies and our results is that more efficient INPs were observed during the bloom conditions, as they nucleate ice at warmer temperatures (i.e., > -15°C), a situation not observed in our study.

- Centrifugal pumps can be harsh on biology and this could be affecting overall INP concentrations. Have authors considered this and measured if the number of INP at specific temperature changes over time? Another option is to perform experiments with other pumps to test affect on INP concentration.

    A/ Thank you for your comment and suggestion. We agree with the reviewer that this could impact the results; however, there it is not a clear consensus in the community. For example, Lee et al. (2015) discussed how pump-induced cell lysis could inhibit the phytoplankton growth. However, in their mesocosm experiments phytoplankton growth was stimulated using a growth medium until the Chl-a threshold was reached. At that point, the plunging waterfall was activated to generate SSA. Even after the operation of the mechanical pump for SSA generation, the authors observed that some phytoplankton species remained alive and continued blooming, as indicated by the increase in chlorophyll concentrations for several days after aerosol generation. On the other hand, intact cells are not necessarily required to act as INPs, as cell fragments or their exudates can also exhibit ice nucleation abilities as shown by Wilson et al. (2015).

    In conclusion, although the usage of the centrifugal pump may have impacted the INP concentrations, unfortunately we did no evaluate its influence. However, we will try to check this out with the UNAM-MARAT in a follow up study where marine biology will be carefully monitored.

    To acknowledge this, the following text was added to the revised manuscript:

    **Lines 644-646:** "However, it is important to note that the usage of a centrifugal pump could impact the marine microorganisms present in the natural seawater samples, potentially affecting their ice nucleating abilities".

Technical comments:

- Line 74: Delete "in."

    A/ Deleted

- Line 120: Suggest using "species" instead of "particles" as most particles can contain more than just a diatom or dinoflagellate.

    A/ Corrected

- Line 190: Water flow or air flow?

    A/ In this case, the flowmeter measured the water flow. No change was applied.

- Figure 1: This is meticulous, but only 8 screws are accounted for in diagram and text states that 10 are used to hold lid.

    A/ Good eyes, thank you. Figure 1 was corrected in the revised manuscript.

- Line 275: Section 2.4 label

    A/ Corrected

- Line 334: *produced

    A/ Thank you. This was fixed.

- Line 481: Should be decreases, not increases

    A/ Thank you. The word was changed.

- Line 613: *tank

    A/ Thank you. This was fixed.

**References**

DeMott, P. J., Hill, T. C. J., McCluskey, C. S., Prather, K. A., Collins, D. B., Sullivan, R. C., Ruppel, M. J., Mason, R. H., Irish, V. E., Lee, T., Hwang, C. Y., Rhee, T. S., Snider, J. R., McMeeking, G. R., Dhaniyala, S., Lewis, E. R., Wentzell, J. J. B., Abbatt, J., Lee, C., Sultana, C. M., Ault, A. P., Axson, J. L., Diaz Martinez, M., Venero, I., Santos-Figueroa, G., Stokes, M. D., Deane, G. B., Mayol-Bracero, O. L., Grassian, V. H., Bertram, T. H., Bertram, A. K., Moffett, B. F., and Franc, G. D.: Sea spray aerosol as a unique source of ice nucleating particles, Proc. Natl. Acad. Sci., 113, 5797–5803, https://doi.org/10.1073/pnas.1514034112, 2016.

Lee, C., Sultana, C. M., Collins, D. B., Santander, M. V., Axson, J. L., Malfatti, F., Cornwell, G. C., Grandquist, J. R., Deane, G. B., Stokes, M. D., Azam, F., Grassian, V. H., and Prather, K. A.: Advancing Model Systems for Fundamental Laboratory Studies of Sea Spray Aerosol Using the Microbial Loop, J. Phys. Chem. A, 119, 8860–8870, https://doi.org/10.1021/acs.jpca.5b03488, 2015.

McCluskey, C. S., Hill, T. C. J., Malfatti, F., Sultana, C. M., Lee, C., Santander, M. V., Beall, C. M., Moore, K. A., Cornwell, G. C., Collins, D. B., Prather, K. A., Jayarathne, T., Stone, E. A., Azam, F., Kreidenweis, S. M., and DeMott, P. J.: A dynamic link between ice nucleating particles released in nascent sea spray aerosol and oceanic biological activity during two mesocosm experiments, J. Atmos. Sci., 74, 151–166, https://doi.org/10.1175/JAS-D-16-0087.1, 2017.

Prather, K. A., Bertram, T. H., Grassian, V. H., Deane, G. B., Stokes, M. D., DeMott, P. J., Aluwihare, L. I., Palenik, B. P., Azam, F., Seinfeld, J. H., Moffet, R. C., Molina, M. J., Cappa, C. D., Geiger, F. M., Roberts, G. C., Russell, L. M., Ault, A. P., Baltrusaitis, J., Collins, D. B., Corrigan, C. E., Cuadra-Rodriguez, L. A., Ebben, C. J., Forestieri, S. D., Guasco, T. L., Hersey, S. P., Kim, M. J., Lambert, W. F., Modini, R. L., Mui, W., Pedler, B. E., Ruppel, M. J., Ryder, O. S., Schoepp, N. G., Sullivan, R. C., and Zhao, D.: Bringing the ocean into the laboratory to probe the chemical complexity of sea spray aerosol, Proc. Natl. Acad. Sci. U. S. A., 110, 7550–7555, https://doi.org/10.1073/pnas.1300262110, 2013.

Wilson, T. W., Ladino, L. A., Alpert, P. A., Breckels, M. N., Brooks, I. M., Browse, J., Burrows, S. M., Carslaw, K. S., Huffman, J. A., Judd, C., Kilthau, W. P., Mason, R. H., McFiggans, G., Miller, L. A., Najera, J. J., Polishchuk, E., Rae, S., Schiller, C. L., Si, M., Temprado, J. V., Whale, T. F., Wong, J. P. S., Wurl, O., Yakobi-Hancock, J. D., Abbatt, J. P. D., Aller, J. Y., Bertram, A. K., Knopf, D. A., and Murray, B. J.: A marine biogenic source of atmospheric ice-nucleating particles, Nature, 525, 234–238, https://doi.org/10.1038/nature14986, 2015.

---

## Author Comment (AC2)

We would like to thank both reviewers for their constructive suggestions, which have significantly contributed to improve the quality and clarity of our manuscript. We carefully considered each comment and have made the necessary modifications accordingly. Specific responses to the reviewers' comments, along with the corresponding changes in the manuscript, are provided below in red text.

**Reviewer 2**

The paper describes a newly designed marine aerosol tank used to generate simulated sea spray aerosol in laboratory settings. The authors give a comprehensive overview of other aerosol tanks in the introduction and carefully describe their results in the context of other tank studies. The tank setup is described in detail, including a clear figure. The tank was tested using both artificial seawater and seawater gathered from the Gulf of Mexico. The analysis of experimental results is given, including both chemical and physical characterization of the aerosol generated from the waters. While results are described in the context of other aerosol reference tank studies, the results from this study are not described in clear detail. More quantification should be given, and the authors should be careful to avoid vague language such as stating that "a better correlation" was found rather than giving exact values. With revision, this paper represents an interesting addition to lab-based sea spray aerosol methods.

> A/ We sincerely appreciate the reviewer's detailed and constructive feedback. The revised manuscript provides clearer and a more precise descriptions of our results, including exact values where applicable.

L403: "likely due to the decomposition of organic matter" – without a measurement of the organic content in water at the time of measurement, this cannot be stated definitively, as turbidity could also be due to suspension of inorganic particulate matter.

> A/ We agree with the reviewer. Given that we were unable to measure the organic content, we cannot rule out other possibilities. Considering your suggestion, the following text was added to the revised manuscript:
>
> **Lines 461-462:** "This could be attributed to a combination of organic matter decomposition and suspended inorganic particulate matter"

L443: it needs to be considered in the analysis that water samples sat at room temperature for fourteen days prior to analysis. This will have a major effect on water chemistry since only organisms >50 µm were filtered out. This would essentially remove all grazers and zooplankton from the system while leaving phytoplankton and bacteria to live on the collected nutrients for fourteen days and cannot be considered truly representative of natural water composition. Nitrate, ammonia, and other nutrients will be metabolized by phytoplankton as the water sits, artificially lowering the concentration.

> A/ We thank the reviewer for her/his comment that helps to clarify the context of the samples' treatment. While we acknowledge that the biological activity may have been significantly altered during the transport/storage period, the goal of our study was not to specifically investigate these effects. Our intention was to evaluate the impact of transportation (and storage) from the collection of samples to the time of the actual

SSA generation experiments. We did this because Mexico City (and our laboratory) is far away from the coast, and therefore, most of the seawater samples need to be transported to Mexico City, unless we go with our setup to each sampling spot. Unfortunately, the latter option is very expensive and not completely feasible.

Therefore, the scope of our study was not to quantitatively assess how transportation and storage influenced the chemical and biological composition of the samples. However, we recognize that this as a limitation in our study. To acknowledge this important point, the following text was added to the revised manuscript:

**Line 291**: "The sample was transported and stored at room temperature".

**Lines 464-473**: "Several factors, including nutrient availability, temperature, oxygen levels, light, and predation, determine the survival of microorganisms. The applied filtration may have removed grazers and other zooplanktonic organism, which could have influenced the development of microbial communities and, consequently, affected the aerosol concentrations. However, some studies suggest that certain species can withstand adverse conditions e.g., metabolic activity can slow down at lower temperatures or  certain phytoplankton and bacteria species may persist in the absence of many predators (Chakraborty et al., 2012; Kennedy et al., 2019). Although it was not the scope of the present study, it is important to monitor how the evolution or degradation of biological species present in the seawater samples impact aerosol properties."

L535: Missing a word between the two phytoplankton species

 A/ Thank you. Corrected.

**Line 556**: "*Thalassiosira weissflogii* and *Synechococcus*"

L541: It may also be that the samples used in this study contain a greater proportion of decomposed or dying material than the two studies listed. While this is addressed later in the paper, it should also be considered here.

 A/ Thank you for the suggestion. To provide a better discussion, the following text was added to the revised manuscript.

**Lines 570-572**: "Another possible explanation for the observed differences between the present and former studies is that our samples likely contain a greater proportion of decomposed or dying material due to their transportation from the coast to the laboratory (Section 4.5)".

L557: Please give actual correlation values and discuss statistical significance of given values

 A/ Thank you for the suggestion. We have now included the actual Spearman's correlation coefficients and their respective p-values in the revised manuscript. Additionally, a discussion of the non-statistically significant values has been included.

**Lines 586-595**: "A better correlation was observed in the BoSM samples ($Na^+$ [$\rho = 0.94$, $p = 0.02$], $Cl^-$ [$\rho = 0.88$, $p = 0.03$], $Mg^{2+}$ [$\rho = 0.83$, $p = 0.06$], $Ca^{2+}$ [$\rho = 0.20$, p

= 0.71], and $NO_3^-$ [$\rho$ = -0.08, p = 0.92]) from April 9 and $Na^+$ [$\rho$ = 0.84, p = 0.04], $Ca^{2+}$ [$\rho$ = 0.84, p = 0.04], $Mg^{2+}$ [$\rho$ = 0.81, p = 0.07], $NO_3^-$ [$\rho$ = -0.84, p = 0.04], and $Cl^-$ [$\rho$ = -0.08, p = 0.87] from April 25) than in the PoV sample ($Na^+$ [$\rho$ = 0.58, p = 0.24], $NO_3^-$ [$\rho$ = 0.55, p = 0.27], $Mg^{2+}$ [$\rho$ = 0.46, p = 0.37], $Cl^-$ [$\rho$ = 0.46, p = 0.37], and $Ca^{2+}$ [$\rho$ = -0.03, p = 0.98]). While $Mg^{2+}$ showed a relatively high correlation in the BoSM samples, it did not reach the threshold for statistical significance ($p < 0.05$). This suggests that, while $Mg^{2+}$ may be present in SSA, its role in ice nucleation remains uncertain. Therefore, further research is needed to determine if $Mg^{2+}$ is a key driver in ice formation in marine environments".

L565-582: This should be considered earlier in the paper as impacts are likely to impact all results given.

A/ Thank you for your suggestion. Although this was clearly stated in Section 2.4 in the original manuscript, for clarity this acknowledge in **Lines 570-572** in revised manuscript.

**References**

Chakraborty, S., Bhattacharya, S., Feudel, U., and Chattopadhyay, J.: The role of avoidance by zooplankton for survival and dominance of toxic phytoplankton, Ecol. Complex., 11, 144–153, https://doi.org/10.1016/j.ecocom.2012.05.006, 2012.

Kennedy, F., Martin, A., Bowman, J. P., Wilson, R., and McMinn, A.: Dark metabolism: a molecular insight into how the Antarctic sea-ice diatom Fragilariopsis cylindrus survives long-term darkness, New Phytol., 223, 675–691, https://doi.org/10.1111/nph.15843, 2019.